# Divergent Heat Stress Responses in *Bactrocera tryoni* and *Ceratitis capitata*

**DOI:** 10.3390/insects15100759

**Published:** 2024-09-30

**Authors:** Kay Anantanawat, Alexie Papanicolaou, Kelly Hill, Yalin Liao, Wei Xu

**Affiliations:** 1Food Futures Institute, Murdoch University, Perth, WA 6150, Australia; 2Hawkesbury Institute for the Environment, Western Sydney University, Richmond, VA 2753, Australia; a.papanicolaou@westernsydney.edu.au; 3South Australian Research and Development Institute, Hartley Grove, Urrbrae, SA 5064, Australia; 4Ithree Institute, University of Technology, Sydney, NSW 2007, Australia

**Keywords:** Queensland fruit fly, Mediterranean fruit fly, heat treatment, heat shock protein

## Abstract

**Simple Summary:**

Fruit flies are major pests that cause extensive damage to fruits and vegetables worldwide. Understanding how these insects respond to heat treatment is crucial for developing effective methods to control them. In this study, we investigated how the Queensland fruit fly and the Mediterranean fruit fly react at a molecular level when exposed to heat. We found that each species responds differently, activating different genes to cope with the stress. By identifying the specific genes involved, we can design targeted heat treatments that are more effective for each species. This tailored approach can improve the efficiency of pest control, reduce the need for harmful chemicals, and minimise damage to the environment. Our findings offer valuable insights for developing better strategies to manage fruit fly populations, ensuring healthier crops and reducing economic losses for farmers. This research highlights the importance of understanding species-specific responses to stress and provides a foundation for future studies aimed at optimising pest management techniques.

**Abstract:**

Invasive Tephritid fruit flies rank among the most destructive agricultural and horticultural pests worldwide. Heat treatment is commonly employed as a post-harvest method to exterminate fruit flies in fruits or vegetables. These pest species exhibit distinct tolerance to heat treatments, suggesting that the molecular pathways affected by heat may differ among species. In this study, the Queensland fruit fly (Qfly), *Bactrocera tryoni*, was utilised as a model investigate its molecular response to heat stress through heat bioassays. RNA samples from flies before and after heat treatment were extracted and sequenced to identify genes with significant changes in expression. These findings were compared to another serious Tephritid fruit fly species, the Mediterranean fruit fly (Medfly), *Ceratitis capitata*, under similar heat treatment conditions. The analysis reveals only three common genes: heat shock protein 70 (HSP70), HSP68, and 14-3-3 zeta protein. However, despite these shared genes, their expression patterns differ between Qfly and Medfly. This suggests that these genes might play different roles in the heat responses of each species and could be regulated differently. This study presents the first evidence of differing molecular responses to heat between Qfly and Medfly, potentially linked to their varied origins, habitats, and genetic backgrounds. These findings offer new insights into Tephritid fruit fly responses to heat at the molecular level, which may help refine post-harvest strategies to control these pests in the future.

## 1. Introduction

Tephritid fruit flies are of significant economic importance in agriculture and horticulture, with several species causing enormous damages to fruits, vegetables, and other crops around the world [1]. For example, the Oriental fruit fly, *Bactrocera dorsalis* (Hendel) (Diptera: Tephritidae), is a highly invasive pest that damages tropical fruits, vegetables, and nut crops [2]. The Mediterranean fruit fly, *Ceratitis capitata* (Wiedemann) (Diptera: Tephritidae), also known as Medfly, can infest over 200 fruits and vegetables, resulting in severe destruction and degradation [3]. The Queensland fruit fly (Qfly), *Bactrocera tryoni* (Froggatt) (Diptera: Tephritidae), is native to subtropical coastal Queensland and northern New South Wales in Australia and causes significant damage to Australian fruit crops annually [4]. In recent decades, the ban on several pesticides formerly used to control Qfly has exacerbated its threat, posing challenges to world trade and market access for horticultural products [3]. 

Strict biosecurity policies have been established worldwide to mitigate the risks and damages caused by Tephritid fruit fly. Pre-harvest measures, including the Sterile Insect Technique (SIT), spraying, lures, surveillance, and inspections are widely applied to control fruit fly pest species [5,6]. Various post-harvest treatment approaches are employed, including chemical treatments (e.g., fumigation) and non-chemical treatments (e.g., cold, heat and irradiation) [7,8,9]. Due to public concerns about chemical residues, environmental pollution, and insect resistance, non-chemical techniques such as heat and cold treatments are increasingly preferred for post-harvest control [10,11,12]. It has been reported that heat treatment not only controls insects and pathogens inside fruits and vegetables but also significantly reduces softening, maintains firmness, and extends shelf-life [13]. 

However, there is a knowledge gap regarding the biological response of the Tephritid fruit flies to heat. Questions such as which molecular pathways are involved in killing fruit flies by heat and whether these pathways are the same across different fruit fly species remain unanswered. Investigating the molecular responses of fruit flies to heat is crucial for improving post-harvest treatments. Understanding these responses could illuminate the molecular mechanisms involved and lead to the development of protocols that combine lower-dose stresses, resulting in higher pest mortality while minimising damage to the fruits. 

Next-generation sequencing (NGS) has been used in sequencing genomes and transcriptomes of various Tephritid fruit fly species, providing an invaluable data for exploring their molecular mechanism [14,15,16]. This technology becomes crucial for identifying fruit fly molecular responses to different stressors. For example, 31 candidate genes with significant changes in expression were identified during heat treatment in Medfly [17]. These genes may play critical roles in immunity, cell death, apoptosis, autophagy, cellular response to heat, and protein folding in other animal species [17]. However, it is unknown if these 31 genes played the same roles in other fruit fly species, such as Qfly. 

Medfly originated in Africa and has spread throughout the Mediterranean region, southern Europe, the Middle East, South and Central America, and Western Australia, now inhabiting most tropical and subtropical areas of the world [3]. Qfly is a native to subtropical coastal Queensland and northern New South Wales in Australia, being broadly recognised as one of the world’s most destructive economic pests of the horticultural industry [4]. Besides significantly decreasing fruit production and quality, Qfly severely impacts trade with sensitive local and international markets [4]. Given their different origins and habitats, it is hypothesised that Qfly and Medfly may have different molecular responses to heat treatment. 

A previous study investigated Medfly molecular response to heat, providing valuable data [17]. Here, heat bioassays were performed on Qfly, and RNA samples were extracted and sequenced similarly to identify genes with significant expression changes. The aim was to identify the candidate genes in Qfly involved in the heat response and compare these genes between Qfly and Medfly under heat treatment. 

## 2. Materials and Methods

### 2.1. Fruit Fly Colony

A colony of Queensland fruit fly (Qfly) was originally collected from Ourimbah, NSW, in Australia in 2015 and maintained in the South Australian Research and Development Institute (SARDI) quarantine facility. The Qfly were less than five generations in captivity at shipment to SA and less than 10 generations for this experiment. Each adult cage was lined with voile fabric, and the Qfly adults were provided with yeast and water for feeding. Qfly eggs were collected using egg devices [18], which contain fruit juice with small holes on the side. Female adults laid eggs through the holes, and the eggs were then strained out of the fruit juice and transferred into a liquid diet [19]. The Mediterranean fruit fly (Medfly) colony originated from a laboratory colony at the Department of Primary Industries and Regional Development in Western Australia [17]. Medfly was maintained in SARDI under the same conditions as previously described [17]. Both Qfly and Medfly colonies were reared at 24 °C, 70% humidity, and 12 h light/12 h dark.

### 2.2. Heat Bioassays 

The heat bioassays were conducted using a thermocycler (Kyratec, Mansfield, Australia), as described [17] (Figure 1A). The temperature 44 °C was chosen for this study, as it is standard for Qfly heat treatment [20]. In all experiments, L3 individuals were used for bioassays, as they are reported as the most tolerant stage to various stressors [21]. A sublethal stress dose was used that caused 75% of the population to be unrecoverable, placing them on a mortality trajectory even after returning to room temperature. The experiment included eight exposure periods: 0 (control), 4, 8, 12, 16, 20, 24, 28 and 32 min. A total of 96 larvae were used per replicate for each exposure time, and four replicates were conducted for each timepoint. After heating treatments, all larvae were maintained at 24 °C for observation. If a treated larva failed to pupate, it was considered “dead”. A non-linear regression model was established to determine the dosage causing 75% mortality in treated flies (Figure 1B), which is the timepoint (T1) used to collect RNA samples (Figure 1A). Statistical analysis was performed using R (version 3.6.3) and R Studio (version 1.3.959). Three timepoints were selected to collect RNA samples for sequencing, which are T0 (before treatment or control), T1 (immediately after treatment), and T2 (two hours post-treatment or recovery). A similar heat treatment was performed at 46 °C on Medfly previously [17], a temperature commonly applied in Medfly heat treatments [22].

### 2.3. RNA Extraction and cDNA Library Construction

Qfly L3 individuals were treated using the stress dose shown in Figure 1, which resulted in 75% larval mortality. Individual larvae were frozen in liquid nitrogen immediately following the treatment (T1 timepoint). The remaining larvae were left at 24 °C for two hours to recover before being collected and frozen (T2 timepoint) for RNA extraction. Larve were also collected prior to treatment as controls (T0 timepoint). Each larva collected from T0, T1, and T2 was homogenised in liquid nitrogen, and total RNA was extracted using a GenEluteTM Total RNA Purification Kit (Sigma-Aldrich, Australia). The quality of the extracted RNA samples was assessed using a LabChip (PerkinElmer, USA) at the Australian Cancer Research Facility (ACRF), and all RNA samples showed good quality with RNA integrity numbers (RIN) over 7 (Appendix A). 

Five RNA samples from the same timepoint with similar quality were pooled to create one library, serving as one replicate. Three replicates were prepared for each timepoint, resulting in a total of nine RNA libraries (three T0, three T1, and three T2) (Figure 2). For each library, the total RNA samples were first treated using *DNase* I, and then RNAseq libraries were generated using an adapted protocol from the Peregrine method [23]. The exact same method was used for extracting total RNA and construct cDNA libraries from Medfly L3 [17]. 

### 2.4. Barcoding and Sequencing

First, quantitative real-time polymerase chain reaction (qPCR) was conducted to determine the minimum number of cycles necessary for barcoding, minimising the introduction of duplicates into the reads [17]. The synthesised first-strand cDNA (1 µL) was diluted in 15 µL of nuclease-free water. Then, 4 µL of the diluted cDNA was added to the PCR reaction mix, which included 5 µL of Bio-Rad SsoFast EvaGreen Supermix and 1 µL of 5 µM TS_qPCR/5 uM RT_Hex_qPCR primer mix (Bio-Rad, South Granville, Australia). The qPCR experiment was carried out as follows: 95 °C for 45 s, 30 cycles of 95 °C for 5 s, and 60 °C for 30 s. The optimal cycle number was obtained as described [17] (Appendix A).

Next, 8 µL of first-strand cDNA product was transferred into 9.5 µL of PCR reaction mix containing 2.4 µL of H_2_O, 0.4 µL of 10 mM dNTPs, 4 µL of 5× Phusion buffer, 0.2 µL of Phusion polymerase, and 2.5 µL of each barcode primers (2 µM) (New England Biolabs, Ipswich, MA, USA) (Appendix A). The PCR was carried out as follows: 1 cycle at 80 °C hold and 98 °C for 10 s, followed by the number of cycles obtained from the qPCR optimisation (Appendix A) of 98 °C for 5 s, 58 °C for 10 s, and 72 °C for 20 s, with a final extension at 72 °C for 5 min and a hold at 15 °C.

The PCR products that contain the barcode sequences (Appendix A) were cleaned using 17 µL of Ampure XP beads (Beckman Coulter Genomics, Indianapolis, IN, USA), as described before, and transferred to a magnetic stand to pellet the magnetic beads [17]. After isolation of the beads, the supernatant was discarded. The magnet beads were washed twice using 200 µL of fresh 80% ethanol, resuspended in 20 µL of 10 mM Tris (pH 8.0), and incubated off the magnetic stand for 5 min. Then, the beads were pelleted again using the magnetic stand. Once the solution was clear, the supernatant was transferred into a new microtube and stored at −20 °C. The quantities of the constructed libraries were measured using the Qubit High Sensitivity DNA kit (Thermo Fisher Scientific, Malaga, PER, Australia) and the NEBNext Library Quant Kit (New England Biolabs, Ipswich, MA, USA). Libraries were pooled in equal amounts and then concentrated to the volume required for sequencing using the Illustra GFX PCR DNA and Gel Band Purification Kit (GE Life Science, Marlborough, MA, USA). The final volume required for sequencing was 20 µL of 4 nM pooled libraries.

The quality of the libraries was assessed using a Bioanalyzer (Agilent, Santa Clara, CA, USA) and Qubit (Thermo Fisher Scientific, Scoresby, VIC, Australia). The libraries were then sent to the Ramaciotti Centre for Genomics (Sydney, NSW, Australia) for sequencing on a Nextseq 500 system (read lengths up to 2 × 150 bp). The cDNA libraries were constructed using the same method as previously described [17]. 

### 2.5. RNA Sequencing Data Analysis

The sequencing data generated from Qfly L3 were first processed and trimmed based on base quality using Fastqc and Trimmomatic (Appendix A). Trimmed reads were then aligned with the published *B. dorsalis* genome using DEW software (https://github.com/alpapan/DEW) (accessed on 29 June 2018).

The raw counts from the alignment were analysed using edgeR [24]. Data filtering removed sequences represented by ≤7 reads and genes that were only present in one library. These reads were normalised using edgeR TMM normalisation and used as an effective library/effective count. The edgeR protocol was applied to examine the clustering of the data using the multidimensional scaling (MDS) plot. The variances of the libraries were calculated, and the data were fitted into a general linear model (GLM). Genes that exhibited significant different expression (2-fold change; *p* < 0.05) at T1 and T2 were identified compared to the control (T0). The same method was utilised to analyse RNA sequencing data from Medfly in a previous study [17] to identify Medfly candidate genes potentially involved in the molecular responses to heat. These identified genes were then further compared by using sequencing alignment between Qfly and Medfly to investigate if orthologue genes are detected in both species. 

## 3. Results

### 3.1. Heat Bioassays

The heat bioassay (Figure 1A) results showed that at 44.0 °C, the exposure time required to kill 75% of Qfly L3 larvae was 16 min and 41 s (Figure 1B). This exposure time, referred to as T1, was used in subsequent treatments to collect flies for RNA extraction to study gene expression changes. It was observed that at shorter heat treatment periods (less than 15 min), the values of the log proportion [log(P)] of dead larvae were more dispersed, indicating that Qfly L3 individuals may exhibit more diverse responses to low-dose heat stressors. However, when the treatment period exceeded 15 min, the values of the log proportion [log(P)] of dead larvae became more concentrated, suggesting that Qfly L3 individuals respond more consistently to higher heat stress. These observations suggest that the duration of heat treatment influences the responses of Qfly L3 larvae to heat stress, with shorter exposures leading to more variable responses and longer exposures resulting in more consistent responses. 

### 3.2. Expression Analysis

To identify the candidate genes involved in Qfly responses to the heat treatment, various bioinformatics analyses were performed. The multidimensional scaling (MDS) plot results showed that under heat treatment, the three T0 (before treatment) samples (T0-1, T0-2, and T0-3), three T1 (immediately after treatment) samples (T1-1, T1-2, and T1-3), and three T2 (two hours after treatment) samples (T2-1, T2-2, and T2-3) clustered into three distinct groups (T0, T1, and T2), though some overlap was observed among these groups (Figure 2). 

Differential expression analysis identified the candidate genes that present significant changes in expression (Appendix A). Compared to T0, 8 genes were significantly down-regulated and 25 genes were significant at T1 (Appendix A). At T2, 3 genes were significantly up-regulated, and 20 genes were significantly down-regulated (Appendix A). All these genes are listed in Appendix A. When comparing T1 to T0, the top three up-regulated genes were epidermal growth factor receptor kinase substrate 8-like protein (3554.091×), embryonic lethal abnormal vision (ELAV)-like protein 2 (2469.784×), and neural precursor cell expressed developmentally down-regulated (NEDD) protein 8 (1995.807×). The top down-regulated genes (0.001×) included high-mobility group protein DSP1, mediator of RNA polymerase II transcription subunit 8, activating transcription factor 3, zinc finger protein 729-like, and unconventional myosin-XV. Comparing T2 to T0, the top down-regulated gene was ELAV-like protein 2 (0.0002×). Interestingly, all three up-regulated genes were heat shock proteins (HSPs): HSP68, HSP68-like, and HSP70. HSP68 showed a 10.520-fold increase, HSP68-like showed a 12.681-fold increase, and HSP70 showed a 44.780-fold increase. 

Compared to T0, 8 genes were down-regulated at T1, while 20 genes were down-regulated at T2, with no common genes between these two time points (Figure 3A). Similarly, 25 genes up-regulated their expression at T1, while 3 genes up-regulated at T2, with no common genes between these time points either (Figure 3B). Interestingly, 17 genes up-regulated their expression levels at T1 but down-regulated at T2 (Figure 3C). For example, epidermal growth factor receptor kinase substrate 8-like protein 2 up-regulated its expression level (3554.091×) at T1 but down-regulated (0.0011×) at T2. Similarly, ELAV-like protein 2 up-regulated its expression (2469.784×) at T1 but down-regulated (0.0002×) at T2. No gene was detected that was down-regulated at T1 and then up-regulated at T2.

For the heat treatment on Medfly, we used data from a previous study [17] and reanalyzed it in the same way as in this study. Compared to T0, seven genes were significantly down-regulated and eight genes were significantly up-regulated at T1 (Appendix A). At T2, 30 genes were significantly up-regulated, and 38 genes were significantly down-regulated (Appendix A). When comparing T1 to T0, the top three up-regulated genes were thiamin pyrophosphokinase 1 isoform X1 (600.381×), FAM92A1 isoform X1 (548.808×), and uncharacterised protein LOC101459691 (461.376×). The top down-regulated genes included transmembrane 147 (0.001×), tigger transposable element-derived 1-like (0.003×), and NPC2 homolog (0.038×). Three heat shock proteins, HSP68, HSP70, and HSP27, exhibited down-regulation at T1. Comparing T2 to T0, the top down-regulated gene was trypsin theta-like (0.001×), and the top up-regulated gene was thiamin pyrophosphokinase 1 isoform X1 (767.884×). Interestingly, heat shock proteins including HSP68, HSP23-like, HSP27, and HSP70 showed down-regulation, while HSP83 showed up-regulation. 

### 3.3. Differentially Expressed Genes between Qfly and Medfly after Heat Treatments

We further investigated whether heat treatment elicits similar or different response between Medfly and Qfly [17]. We compared the identified genes between two fly species during their heat treatments and found only three orthologous genes in both: HSP68, HSP70, and a third gene described as 14-3-3 zeta in Medfly but as a high-mobility group protein DSP1 in Qfly. Beside these three genes, no other orthologous genes were identified between Medfly and Qfly during heat bioassays.

Further, we compared gene expression regulation in these two fly species at T1 and T2 separately. As shown in Figure 4, no genes were regulated in the same way (up-regulated or down-regulated) by heat treatment in both Medfly and Qfly. For example, HSP70 was down-regulated at both T1 (0.40×) and T2 (0.63×) in Medfly, whereas HSP70 was up-regulated at T2 (44.783×) in Qfly but showed no significant changes in expression levels at T1. Similarly, 14-3-3 zeta protein was slightly up-regulated at T1 (1.04×) and significantly down-regulated (0.76×) at T2 in Medfly. However, the same gene (high mobility group protein DSP1) showed significant down-regulation (0.001×) at T1 but did not show significant expression change at T2 (Appendix A). These findings suggest that the molecular responses to heat stress are quite different between Qfly and Medfly, indicating species-specific responses to heat stress in these two fruit fly species. 

### 3.4. Heat Shock Proteins (HSPs)

HSPs have been reported with multi-functions in autophagy cell death, cellular response to heat, immunity, protein folding, and apoptosis in various animal species, so they are the target proteins in animal responses to various stressors [25,26]. From RNA sequencing data of both fly species, six HSPs were identified in response to heat treatment. They were HSP23-like, HSP27, HSP68, HSP68-like, HSP70, and HSP83. We summarise their expression changes in Figure 4E. In Medfly L3 heat treatment, three HSP genes (HSP27, -68, and -70) were down-regulated in T1. Four HSP genes (HSP27, -68, -70 and HSP23-like) were down-regulated, while HSP83 was up-regulated in T2. In Qfly L3 heat treatment, no HSP was identified from T1, but three HSP genes (HSP68, HSP70, and HSP68-like) were up-regulated in T2. These results suggest that the HSPs exhibited different expression profiles between two fly species under heat stressors.

## 4. Discussion

This study aimed to investigate the molecular responses of Qfly L3 to heat stressors and compare them to those of Medfly. The candidate genes with significant expression changes during heat treatments were identified, potentially serving as molecular targets to enhance our understanding of how fruit flies respond to stress at the molecular level. By examining gene expression changes in response to heat stress, the molecular mechanisms underlying Qfly’s and Medfly’s responses to heat stressors can be uncovered. These candidate genes could play roles in the flies’ ability to adapt to and survive high-temperature environments. Understanding the molecular basis of these responses could contribute to the development of strategies for controlling fruit fly populations, such as through genetically modified organisms or targeted molecular interventions. 

Furthermore, comparing Qfly and Medfly responses to heat stressors could provide insights into the similarities and differences in their molecular stress responses, which may relate to their evolutionary history, ecological preferences, and physiological characteristics. This information could elucidate the molecular basis of the different thermal tolerance levels observed in various fruit fly species and have implications for pest management strategies and the development of heat-stress-resistant strains of fruit flies for agricultural applications.

Qfly L3 were selected here because they are reported to be the most heat-tolerant stage in fruit flies [21], and Medfly L3 were used in similar studies before [17]. Additionally, more RNA can be extracted from individual L3 larvae than L1, L2, or eggs. In this study, naked L3 larvae were used in heat bioassays, understanding that this differs from how insects are exposed to post-harvest heat in industrial settings, where treatments are directly applied to fruit flies within fruits. The sizes, nutrients compositions, materials, and chemicals of different fruits can influence heat transfer [27], fly development, and infestation methods [28]. Treating fruits and then harvesting fly RNA samples from inside the fruit can result in significant delays, RNA contamination, and degradation. Therefore, naked larvae were used to minimise these risks and focus on the fly responses at the molecular level.

Using the heat bioassay method, a model was built according to the numbers of L3 larvae that developed into pupae after treatment. The treatment period that causes 75% Qfly L3 mortality was determined as the treatment time because it provides significant lethal stress, without resulting in 100% mortality, which could cause RNA degradation.

The MDS plot results of Medfly showed that three T0 samples, three T1 samples, and three T2 samples were clustered into three different groups, respectively [17]. However, here, the MDS plot results of Qfly showed that the three T0 samples, three T1 samples, and three T2 samples clustered into three groups but with some overlap among these groups (Figure 2). For the heat treatment, 44 °C was used as it has been shown to effectively disinfest fruits against Qfly [20]. It is different from the temperature used in a previous study on Medfly [17], which was 46 °C, a temperature commonly applied in Medfly heat treatments [17]. This difference may cause variations in the efficacy of disinfestation between Qfly and Medfly, as the two species may have different thermal tolerances and responses to heat treatment. The lower temperature of 44 °C used for Qfly could be optimised for its physiology, ensuring effective disinfestation without compromising fruit quality. In contrast, the higher temperature of 46 °C applied in Medfly treatments might reflect the species’ resistance to heat or a need for higher temperatures to achieve the same level of disinfestation. Further studies comparing these temperature thresholds could help refine species-specific treatment protocols, minimising fruit damage while ensuring effective pest control. Qfly may have evolved distinct genetic characteristics, such as differential expression of heat-shock proteins or other molecular mechanisms, influencing its response to heat. Additionally, environmental factors, such as the climate where Qfly populations are found, could shape their adaptation to lower heat thresholds, making 44 °C an effective treatment. Furthermore, Qfly’s ability to induce phenotypic plasticity in response to heat stress could contribute to the variability observed in treatment outcomes, as the species may adjust its physiological responses based on the exposure to varying temperatures. These factors highlight the importance of species-specific approaches to heat treatments for effective disinfestation.

A previous study on the molecular response of Medfly to heat provided RNA sequencing data for comparative analysis between Qfly and Medfly [17]. Interestingly, three HSPs, namely, HSP68-like, HSP68, and HSP70, were identified in both studies. HSPs have demonstrated multiple functions in various animal species, including cellular response to heat, autophagy cell death, immunity, protein folding, and apoptosis. HSP20, HSP40, HSP70, and HSP90 in *Liriomyza sativa* and *Liriomyza huidobrensis* can be induced by both heat and cold stresses [29,30]. In our previous studies on Medfly heat and cold treatment, HSP60, HSP70, HSP23-like, and HSP83-like showed significant expression changes [31]. These results suggest that HSPs play critical roles in fruit fly responses to heat and cold stressors. These HSPs and other identified genes can be further functionally studied using RNAi or CRISPR technologies in Qfly. By knocking down or knocking out the candidate genes, the flies’ tolerance to heat stress can be assessed.

We previously performed the same heat treatment experiment with Medfly [17]. When comparing the identified genes and their expression changes between Qfly and Medfly, the results showed that not a single gene exhibited the same expression pattern under heat treatment in both species. This suggests that different fruit fly species respond to heat treatment through different mechanisms, with different genes being activated.

On the other hand, the different molecular responses of the two fruit fly species to heat treatment may be due to selection and local adaptation to their different thermal environments. The Medfly colony used in this study originated from Western Australia, a Mediterranean climate, whereas the Qfly used is native to subtropical coastal Queensland and northern New South Wales in Australia. Their different habitats may lead to their different molecular response to heat.

This study suggests that each fruit fly species may have specific molecular pathways to respond to heat or other stressors. Therefore, heat treatments should be specifically developed for each species rather than applying a one-size-fits-all protocol for fruit fly pest management. This information can be used to design heat treatments that target the identified pathways or genes, aiming to enhance or suppress their expression in a species-specific manner. For example, for genes upregulated in Qfly but downregulated in Medfly in response to heat stress, treatments can be tailored to enhance these genes in Qfly, while suppressing them in Medfly. This could be achieved by adjusting the temperature, duration, and timing of heat treatments to optimise gene expression in each species. Tailoring heat treatments based on species-specific molecular responses can improve the process in several ways: (1) Precision: By considering the specific genes or pathways differentially expressed in response to heat stress in each species, heat treatments can be designed with precision, targeting the molecular mechanisms for each species. (2) Efficiency: Optimising heat treatments based on species-specific molecular responses can result in more efficient treatments, as the temperature, duration, and timing can be tailored to achieve desired effects on gene expression. This can lead to more effective and faster elimination of fruit fly populations, potentially reducing the need for repeated treatments or higher temperatures that may negative impact fruit quality or the environment. (3) Selectivity: Tailored heat treatments can make the process more selective, specifically targeting pest species while minimising impacts on non-target organisms. By exploiting the differential gene expression responses of different fruit fly species to heat stress, treatments can be designed to selectively affect the target species, reducing potential collateral damage to beneficial insects or other non-target organisms.

Tailoring heat treatments specifically for each fruit fly species based on differential expression analysis can improve the process by increasing precision, efficiency, and selectivity, potentially leading to more effective pest management strategies. However, it is important to carefully consider the potential impacts on the environment, fruit quality, and other non-target organisms, as well as to conduct further research and validation to ensure the safety and effectiveness of such tailored treatments. 

Further, the candidate genes identified in Qfly heat treatment may serve as marker genes to evaluate whether a fly has undergone a heat treatment, helping to determine if appropriate heat treatment has been applied. For example, if an Qfly is detected in heat-treated fruits or vegetables, it can be difficult to determine if the fly is a “survivor” of the heat treatment or an accidental intruder that did not receive the treatment. Marker genes could help examine the fly’s previous exposure to heat treatment based on gene expression levels. More studies are needed to investigate these Qfly candidate genes and optimise the approach, which will benefit current fruit fly post-harvest treatment and management.

The findings of this study are particularly relevant in the context of climate change and global warming. As global temperatures continue to rise, the frequency and intensity of heatwaves are expected to increase, posing significant challenges to agricultural systems worldwide. Fruit flies, including the Qfly and Medfly, are major agricultural pests that threaten fruit production and quality. Understanding the molecular responses of these species to heat stress is crucial for developing effective pest management strategies in a warming climate. The differential gene expression patterns observed in this study highlight the potential for species-specific adaptations to changing thermal environments. These insights can inform the design of targeted heat treatments to manage fruit fly populations more effectively, reducing the reliance on chemical pesticides and minimising their environmental impact. Moreover, the identification of key genes involved in heat stress responses could lead to the development of genetically modified strains with enhanced thermal tolerance, ensuring the sustainability of fruit production in the face of climate change. This research underscores the importance of integrating molecular biology with ecological and environmental studies to address the multifaceted challenges posed by global warming.

## Figures and Tables

**Figure 1 insects-15-00759-f001:**
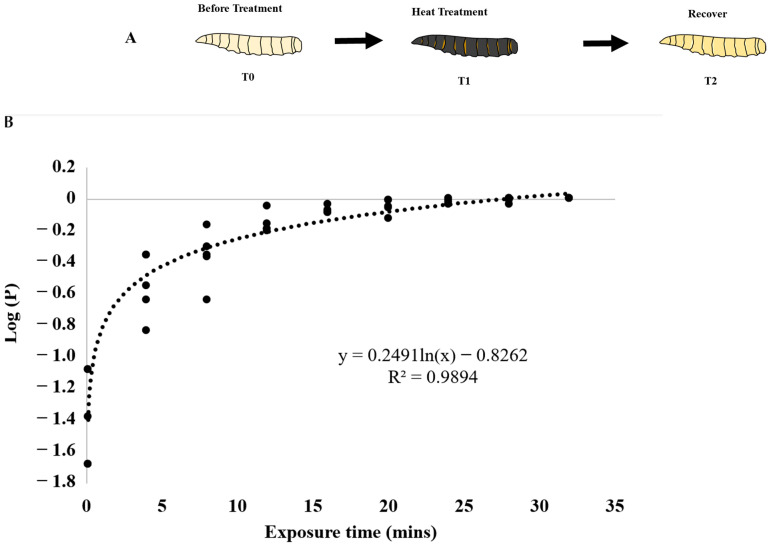
The heat bioassay on *Bactrocera tryoni* L3 individuals. (**A**) The schematic process for the heat treatment on *B. tryoni* L3. (**B**) The proportion of dead *B. tryoni* L3 under 44 °C. In the plot, the black dash-line represents the best fit curve. The Y axis stands for the log proportion [log(P)] of dead larvae to the heat treatment. The results showed that under 44 °C heat treatment, exposure time to kill 75% *B. tryoni* L3 was 16 min 41 s. Three timepoints, namely, T0 (before the treatment), T1 (immediately after the treatment), and T2 (2 h after the treatment), were used. The fly larvae were collected at T0, T1, and T2 timepoints for RNA extraction and sequencing.

**Figure 2 insects-15-00759-f002:**
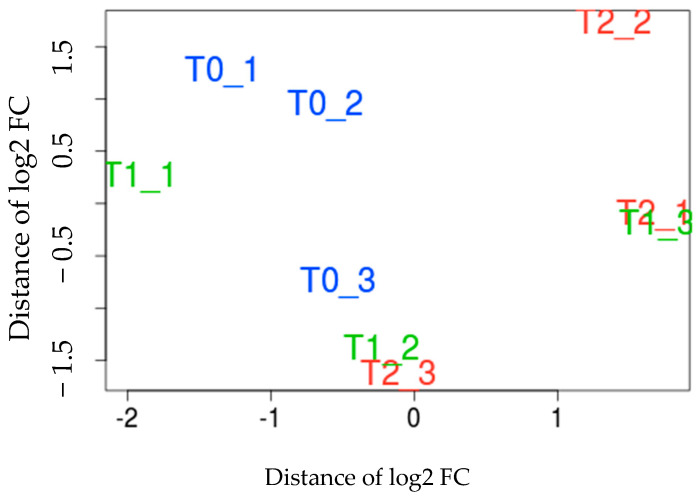
The multidimensional scaling (MDS) of *Bactrocera tryoni* L3 RNA samples at T0, T1, and T2 under heat treatments. MDS plot of the heat treatment samples based on the RNA sequencing data. T0 = before the treatment, T1 = immediately right after the treatment, T2 = 2 h after the treatment; T0_1, T0_2, and T0_3 represent three biological replicates. Log2 FC means Log2 fold change.

**Figure 3 insects-15-00759-f003:**
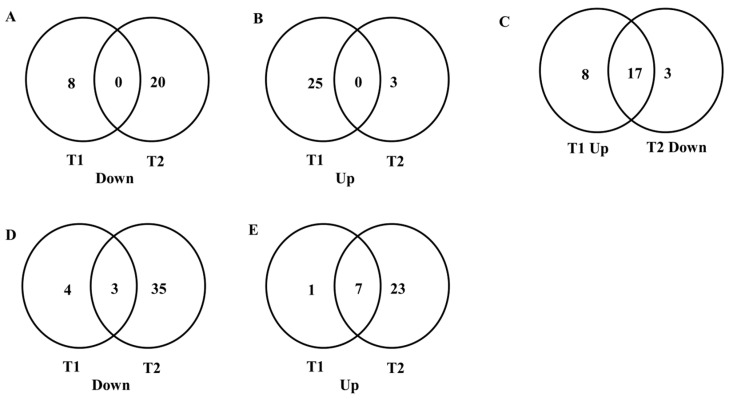
Venn diagrams comparing differentially regulated genes between T1 and T2 in *Bactrocera tryoni* and *Ceratitis capitata* L3 after heat treatments. (**A**) The numbers of genes that were down-regulated at both T1 and T2 after the heat treatment in *B. tryoni*; (**B**) the numbers of genes that were up-regulated at both T1 and T2 after the heat treatment in *B. tryoni*; (**C**) the numbers of genes that were up-regulated at T1 but down-regulated at T2 after the heat treatment in *B. tryoni*; (**D**) the numbers of genes that were down-regulated at both T1 and T2 after the heat treatment in *C. capitata*; (**E**) the numbers of genes that were up-regulated at both T1 and T2 after the heat treatment in *C. capitata*.

**Figure 4 insects-15-00759-f004:**
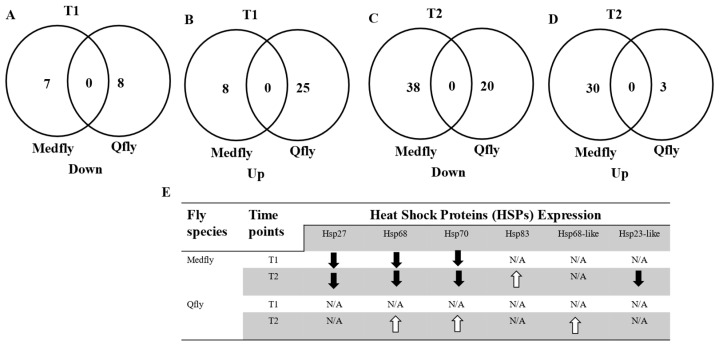
Venn diagrams comparing differentially regulated genes between *Bactrocera tryoni* and *Ceratitis capitata* L3 after the heat treatment. (**A**) The number of genes that were down-regulated at T1 after the heat treatments; (**B**) the number of genes that were up-regulated at T1 after the heat treatments; (**C**) the numbers of genes that were down-regulated at T2 after the heat treatments; (**D**) the number of genes that were down-regulated at T2 after the heat treatments. (**E**) Regulation of heat shock proteins in *B. tryoni* and *C. capitata* L3 individuals responding to stress treatments. The white up arrow represents the up-regulation of the gene, while the black down arrow represents the down-regulation of the gene. N/A indicates that the expression level did not detect significant change in response to stress treatment.

## Data Availability

The datasets during and/or analysed during the current study are available from the corresponding author on reasonable request.

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
