# Peer review of "Divergent Heat Stress Responses in Bactrocera tryoni and Ceratitis capitata"

_insects, 2024, doi:10.3390/insects15100759_

Round 1

Reviewer 1 Report

Comments and Suggestions for Authors

This manuscript presents crucial new insights into the molecular responses of two quarantine fruit fly species to heat. These findings have the potential to significantly advance our understanding and control of these pests in the future.

I think you should include the names of the species in the title and remove the keywords.

Several statements in the introduction lack citations. The authors must provide references for each statement to support their claims.

The first time a species is cited in the text, the authority, order, and family should be included.

In the methodology, the authors must provide detailed information about the laboratory colonies of the two species studied. This includes their origin, maintenance conditions, and any potential genetic variations. This level of detail is crucial for ensuring the transparency and reproducibility of scientific research.

The authors should present how RNA Extraction and cDNA Library Construction were performed for Ceratitis capitata L3 individuals.

No abbreviations should be used in the captions.

In the discussion,  I do not consider it appropriate to compare very distant animals phylogenetically, such as comparing the results obtained with fruit flies with fish.

It's crucial that scientific names are italicized in bibliographic references. This is a standard formatting requirement in scientific manuscripts and helps to distinguish them clearly from other texts.

Although the research does not focus on global warming, the authors should address how their results could be used in studies. This information would further increase the audience's interest.

I have attached a file with more suggestions to enhance the impact and clarity of your manuscript.

Author Response

Thank you for your insightful comments. Our responses to are below:

Reviewer 1

Comments and Suggestions for Authors

This manuscript presents crucial new insights into the molecular responses of two quarantine fruit fly species to heat. These findings have the potential to significantly advance our understanding and control of these pests in the future.

  1. I think you should include the names of the species in the title and remove the keywords.

Done!

  1. Several statements in the introduction lack citations. The authors must provide references for each statement to support their claims.

Done! References were added.

  1. The first time a species is cited in the text, the authority, order, and family should be included.

Done!

  1. In the methodology, the authors must provide detailed information about the laboratory colonies of the two species studied. This includes their origin, maintenance conditions, and any potential genetic variations. This level of detail is crucial for ensuring the transparency and reproducibility of scientific research.

Done!

  1. The authors should present how RNA Extraction and cDNA Library Construction were performed for Ceratitis capitata L3 individuals.

Added

  1. No abbreviations should be used in the captions. Done!
  2. In the discussion,  I do not consider it appropriate to compare very distant animals phylogenetically, such as comparing the results obtained with fruit flies with fish.

The discussion part comparing the results between fruit flies and fish were deleted.

  1. It's crucial that scientific names are italicized in bibliographic references. This is a standard formatting requirement in scientific manuscripts and helps to distinguish them clearly from other texts.

Done!

  1. Although the research does not focus on global warming, the authors should address how their results could be used in studies. This information would further increase the audience's interest.

One paragraph was added discussing the global warming and this study.

  1. I have attached a file with more suggestions to enhance the impact and clarity of your manuscript.

Thanks. We have revised according 

Reviewer 2 Report

Comments and Suggestions for Authors

In this manuscript, the authors studied the heat stress response in the Queensland fruit fly (Qfly) and compared this new data set with a previous data set from the Mediterranean fruit fly (Medfly). The authors found a differential response between the QFly and the MedFly after heat treatment, indicating specific molecular responses between fruit flies. While the authors show a suitable strategy to evidence different molecular responses to heat between both species, some aspects need to be addressed to improve the paper.  

Major concern:

The fundamental major flaw of this work is in figure 2. My recommendations are the next:

1) The multidimensional scaling (MDS) analysis shows a high degree of variation between samples, indicating a loss of apparent clustering between conditions. In Line 217, the authors indicated that some samples overlap, and in the results, they indicated that replicates that do not cluster were removed from the analysis. I also recommend indicating in the figure legend which replicate was removed for the analysis. Otherwise, readers might speculate about whether replicate three was removed or which one was removed.

2) Lines 226-238. The authors show two volcano plots in Figure 2. Down-regulated genes are indicated in pink circles, and up-regulated genes in blue squares. Both plots show more than three up-regulated genes (more than three blue squares). However, the text indicates, Comparing to T0, eight genes were significantly down-regulated, and 25 genes were significantly at T1 (Fig. 2B).”

2.1 recommend correcting the text according to the figure because both volcano plots show more than three blue squares.

2.2 I also recommend adding the letter “C” to the figure panel and explaining better in the main text and figure legend which volcano plot corresponds to T0 vs T1 and which one to T0 vs T2.

2.3 The discussion will benefit if the authors discuss the variability observed between the three biological replicates because, in fact, it is something that was not observed in the work used to compare this data set (reference 13). Is the difference in temperature used in the heat treatment in Qfly (44°C) vs Medfly (46°C) related to such variability?, Can such variability be due to specific genetic or environmental characteristics in Qfly? Do the authors know if Qfly is able to induce a plasticity response to heat reflected in the sample variability?

Minor comments:

Figure 1A needs to be mentioned in the text in the result section. My recommendation is to indicate “(Fig 1A)” after the words “The heat bioassays” in line 197.

The introduction reads well. The authors state the need to establish the heat response at the molecular level between fruit flies of agricultural importance worldwide. However, some big paragraphs need references. Add a reference for the next lines:

Lines 80-82-add a reference for “Medfly originated in Africa and has spread throughout the Mediterranean region, southern Europe, the Middle East, South and Central America, and Western Australia, now inhabiting most tropical and subtropical areas of the world”

Lines 82-84-add a reference for “Qfly is a native to subtropical coastal Queensland and northern New South Wales in Australia, which is broadly recognized as one of the world’s most destructive economic pests of horticultural industry”

Lines 280-282-add a reference for “HSPs have been reported with multi-functions in autophagy cell death, cellular response to heat, immunity, protein folding and apoptosis in various animal species, so they are the target proteins in animal responses to various stressors.

Comments on the Quality of English Language

Minor comments: 

Line 210. Check the misspelling word “points” in figure legend 1.

Lines 315-316. Correct misspelling word “influent”.

Author Response

Thank you so much for your insightful comments and our responses are below:

In this manuscript, the authors studied the heat stress response in the Queensland fruit fly (Qfly) and compared this new data set with a previous data set from the Mediterranean fruit fly (Medfly). The authors found a differential response between the QFly and the MedFly after heat treatment, indicating specific molecular responses between fruit flies. While the authors show a suitable strategy to evidence different molecular responses to heat between both species, some aspects need to be addressed to improve the paper.  

Major concern:

The fundamental major flaw of this work is in figure 2. My recommendations are the next:

  • The multidimensional scaling (MDS) analysis shows a high degree of variation between samples, indicating a loss of apparent clustering between conditions. In Line 217, the authors indicated that some samples overlap, and in the results, they indicated that replicates that do not cluster were removed from the analysis. I also recommend indicating in the figure legend which replicate was removed for the analysis. Otherwise, readers might speculate about whether replicate three was removed or which one was removed.

Thanks. This sentence was deleted. We went through the detailed analysis to check the method part and found actually we did not remove any replicates in this study.

2) Lines 226-238. The authors show two volcano plots in Figure 2. Down-regulated genes are indicated in pink circles, and up-regulated genes in blue squares. Both plots show more than three up-regulated genes (more than three blue squares). However, the text indicates, Comparing to T0, eight genes were significantly down-regulated, and 25 genes were significantly at T1 (Fig. 2B).”

2.1 I recommend correcting the text according to the figure because both volcano plots show more than three blue squares.

Thank you for that. We removed the volcano plots because this figure used a different cutoff value. Therefore, it is not consistent with our tables.

2.2 I also recommend adding the letter “C” to the figure panel and explaining better in the main text and figure legend which volcano plot corresponds to T0 vs T1 and which one to T0 vs T2.

             The volcano plot was deleted as above.

2.3 The discussion will benefit if the authors discuss the variability observed between the three biological replicates because, in fact, it is something that was not observed in the work used to compare this data set (reference 13). Is the difference in temperature used in the heat treatment in Qfly (44°C) vs Medfly (46°C) related to such variability?, Can such variability be due to specific genetic or environmental characteristics in Qfly? Do the authors know if Qfly is able to induce a plasticity response to heat reflected in the sample variability?

 One new paragraph was added to discuss these questions.

Minor comments:

Figure 1A needs to be mentioned in the text in the result section. My recommendation is to indicate “(Fig 1A)” after the words “The heat bioassays” in line 197.

Done!

The introduction reads well. The authors state the need to establish the heat response at the molecular level between fruit flies of agricultural importance worldwide. However, some big paragraphs need references. Add a reference for the next lines:

Lines 80-82-add a reference for “Medfly originated in Africa and has spread throughout the Mediterranean region, southern Europe, the Middle East, South and Central America, and Western Australia, now inhabiting most tropical and subtropical areas of the world”

Done!

Lines 82-84-add a reference for “Qfly is a native to subtropical coastal Queensland and northern New South Wales in Australia, which is broadly recognized as one of the world’s most destructive economic pests of horticultural industry”

Done!

Lines 280-282-add a reference for “HSPs have been reported with multi-functions in autophagy cell death, cellular response to heat, immunity, protein folding and apoptosis in various animal species, so they are the target proteins in animal responses to various stressors.

 Done!

Minor comments: 

Line 210. Check the misspelling word “points” in figure legend 1.

Done!

Lines 315-316. Correct misspelling word “influent”.

Done!
